# Habitat Enrichment Causes Changes in Fish Behavioural Characteristics: A Case Study of *Sparus latus*

**DOI:** 10.3390/biology13060364

**Published:** 2024-05-22

**Authors:** Yu Guo, Zhanlong Chen, Chuanxin Qin, Gang Yu, Jia Zhang

**Affiliations:** 1South China Sea Fisheries Research Institute, Chinese Academy of Fishery Sciences, Key Laboratory of Marine Ranching, Ministry of Agriculture and Rural Affairs, Guangzhou 510300, China; guoyu25895177@163.com (Y.G.); 15054035603@163.com (Z.C.); gyu0928@163.com (G.Y.); zj19922812035@163.com (J.Z.); 2National Agricultural Experimental Station for Fishery Resources and Environment Dapeng, Shenzhen 518121, China; 3Hainan Seed Industry Laboratory, Sanya 572025, China

**Keywords:** habitat enrichment, behavioural characteristics, artificial reefs, *Sparus latus*

## Abstract

**Simple Summary:**

Habitat restoration is a key way to restore fishery resources. Artificial reefs, as important habitat restoration measures, have achieved the goal of protecting and increasing fishery resources. However, research on the breeding process and models of artificial reefs for ensuring fishery resources is lacking. In this study, the behavioural strategies of *Sparus latus* (*S. latus*) under the influence of habitat abundance were studied to provide a theoretical basis for restoring the habitat and promoting the proliferation of fishery resources. There was no significant difference in reef first contact time of juvenile *S. latus* during the day or at night. Enrichment structures reshaped the habitat preferences of *S. latus*. An increase in habitat enrichment promoted *S. latus* clustering at night. The reef opening ratio significantly affected the reef-tropism and clustering behaviours of *S. latus*. The light intensity significantly influenced the exploration and activity patterns of *S. latus*.

**Abstract:**

To better understand the habitat preferences and behavioural ecology of *Sparus latus*, we performed an experiment using box-shaped reefs as habitat enrichment materials, allowing us to determine the behavioural strategies and drivers involved in the response to different enrichment structures. The results showed that the first contact time of *S. latus* was negatively correlated (Pearson’s correlation, *p* < 0.005) with the distribution rate in the artificial reef area. Enrichment structures affected the habitat preferences of *S. latus*, and there was a significant difference in the average distribution rate between the control and treatment groups (Adonis, *p* < 0.001). The opening ratio (Adonis, *R*^2^ = 0.36) explained the distribution difference of *S. latus* better than the opening shape (Adonis, *R*^2^ = 0.12). In the absence of an enrichment structure, *S. latus* remained more active during the daytime, exhibiting poor clustering, while in the presence of an enrichment structure, *S. latus* exhibited clustered movement at night. The opening ratio was negatively correlated with the average interindividual distance (Spearman’s correlation, *p* < 0.01) and showed a significant positive correlation with the average distribution rate in the reef area (Spearman’s correlation, *p* < 0.001), indicating that the reef opening ratio significantly affected the reef-tropism and clustering behaviours of *S. latus*. The light intensity was negatively correlated with the average distance moved, and the average speed (Spearman’s correlation, *p* < 0.05) was significantly positively correlated with the reef first contact time (Spearman’s correlation, *p* < 0.001), indicating that the light intensity affected the exploration and activity patterns of *S. latus*. These results provide a research basis for analysing the pattern and process of fish proliferation induced by artificial reef habitats.

## 1. Introduction

Habitat restoration is a key way to restore fishery resources. Incorporating appropriate substrates, shelters, plants, and other structures into the habitat of aquatic organisms for habitat enrichment could enhance their environmental adaptability and increase their release survival rate [1,2]. However, the effect of habitat enrichment is influenced by the studied species, lifestyle, growth stage, and habitat structure [3,4]. Therefore, analysing the behavioural characteristics of fish in artificial reefs in response to habitat environment changes through habitat enrichment is crucial for exploring the effects of artificial reefs on fish release processes and patterns.

Artificial reefs are structures that are placed in natural environments to protect and nurture marine biological resources and constitute a typical application model for habitat enrichment [5,6,7]. In recent years, the deployment of artificial reefs in specific areas to restore fishery resources has been extensively practised both domestically and internationally. *Epinephelus akaara* mostly inhabits the interior of tubular reefs or the narrow crevices between reefs and the shaded areas covered by reefs, while larger reef models can yield greater fish-gathering effects [8]. *Paralichthys olivaceus* prefers to inhabit sandy or rocky areas. After their placement in reefs, they generally live near reefs or enter reefs to avoid enemies [9]. Artificial reef release provides a necessary and safe habitat for reef fish and other marine organisms to gather, feed, reproduce, grow and avoid enemies. Studies have indicated that reef fish species such as *Pagrus major*, *Acanthopagrus schlegelii*, *Plectrhynchus cictus*, and *Sebastes schlegeli* all exhibit significant reef responses and suitable preferences for artificial reefs. Light is an important environmental factor that influences the feeding, growth, reproduction, and behaviour patterns of fish [10,11,12]. When faced with different lighting conditions, fish exhibit phototaxis or photoavoidance behaviour characteristics, which impact their migration and distribution in water. Research on *Pelteobagrus vachelli* has revealed that the suitable light intensity ranges from 0–10 lx [13], indicating that light intensity is another key factor shaping the reef-driving behaviour of fish.

At present, there are relevant studies on the habitat selection behaviour of aquatic organisms [14,15,16]. Different habits of aquatic organisms lead to varying habitat choices. Enhancing the reef-driving behaviour of released fish through behavioural domestication measures has remained a key link in constructing marine ranches [17,18]. The reef-driving behaviour of fish refers to the directional movement caused by their response to external stimuli [19,20]. Reef fish generally exhibit the instinctive behaviour of gathering around reefs, while their reef-driving behaviour is mainly characterized by first contact time [21], distribution rate [22], and related movement indicators [23]. First contact time can be quantified as the time initially needed for fish to approach a novel object, explained as the fear of fish towards new objects. The distribution rate is the ratio of the number of fish observed within a fixed period to the total number of fish in a certain area and can be used to measure the preference of fish for habitat selection. The exercise indicators of the average speed, average distance moved, average turning rate movement, and percentage of the active time can be employed to indicate the activity ability of fish, while the nearest neighbour distance and average interindividual distance indicate the clustering characteristics of fish.

Yellowfin seabream (*Sparus latus*) is a warm-water bottom fish in the shallow sea. It lives in coastal waters and estuaries and prefers to live in reef sea areas. It is a euryhaline and eurytherm species and generally does not travel large distances, so it provides high food and resource conservation values [24]. In recent years, due to overfishing, ecological damage and other factors, the population of *S. latus* in natural waters has decreased, while the released *S. latus* cannot adapt to the changing natural environment, cannot avoid predators, exhibits a low survival rate, and demonstrates a low welfare level [25]. The effects of the proliferation and release of yellowfin snapper failed to meet expectations, while the effect of fishery resource restoration was unsatisfactory. At present, studies on *S. latus* have focused mainly on the fields of biology, artificial and natural breeding, resource assessment, immunity, and feeding [26]; however, there is relatively little research on the application of habitat enrichment in fish release, as well as suitable habitat enrichment conditions. To determine the most suitable habitat enrichment method, we must determine the preferences for habitat selection, activity characteristics, and their interactions, as well as their effects on behaviour development. Therefore, in this study, different types of artificial reefs were adopted as habitat enrichment conditions to explore the effects of the structure and size of artificial reefs on the habitat selection preferences and behavioural strategies of juvenile *S. latus,* and the driving factors of this behaviour were analysed, revealing whether the opening structure and size of artificial reefs are key factors of habitat abundance design. These experiments were performed to improve the survival rate of released fish through behavioural domestication methods such as habitat enrichment and provide a theoretical basis and reference for selecting and designing artificial reefs.

## 2. Materials and Methods

### 2.1. Experimental Animals and Fish Maintenance

We randomly selected 540 unharmed *S. latus* individuals (body length: 15.6 ± 1.4 cm; body height: 6.1 ± 0.7 cm; body weight: 102.8 ± 21.2 g) from the Tropical Aquatic Research Center of the South China Sea Fisheries Research Institute of the Chinese Academy of Fishery Sciences.

Before the experiment, the fish were placed in a temporary breeding pond (200 cm × 120 cm × 80 cm) equipped with mechanical and biological filtration pumps to filter the aquaculture seawater. The pond was maintained, the water environment was monitored and cleaned, and water was changed (1/3 of the total volume) once a week. The experimental seawater temperature was 28.1 ± 0.4 °C, the salinity was 33.0 ‰, the pH was 8.42 ± 0.14, the dissolved oxygen concentration was maintained at 6 mg/L or above, and the tank was temporarily aerated for 24 h. Fifty grams of bait was added at 9 am and 5 pm each day, and after 0.5 h of feeding, the unused bait and faeces in the temporary breeding pond were cleaned using the siphon method.

### 2.2. Experimental Reef Types

The enrichment facility was box-shaped with five identical grey-coloured acrylic plates. Due to the limitations in the size of the experimental pond, the artificial reef size is 30 cm × 30 cm × 30 cm, comprising five identical acrylic plates spliced together, with a thickness of 0.5 cm for each board. Four diamond-, circular-, or square-shaped holes were set on each side of the artificial reef. The size of these openings was designed based on the height multiple of the experimental fish, and the *S. latus* body height was approximately 6.0 cm, so the artificial reef opening heights were set to 3.0 cm, 6.0 cm, 9.0 cm, and 12.0 cm. The opening was 0.5, 1.0, 1.5 and 2 times greater than the experimental fish body height, for a total of 12 artificial reef models (Figure 1).

### 2.3. Experimental Environment and Equipment

The experimental pond (260 cm × 230 cm × 100 cm) was a square groove with a blue background, and the water depth was 60 cm. During the experiment, multiparameter equipment (YSI, Professional Plus, Yellow Springs, OH, USA) was used every day to measure the water conditions, dissolved oxygen concentration (mg/L), pH, temperature (°C), and salinity. In this process, seawater flow was deactivated, the water was not aerated, and the fish were not fed. The area division method was used to mark the placement position of the reef model with a deep colour in the middle (Figure 2). The area surrounding the reef and the internal area of the reef were divided into artificial reef areas (Zone C), and the area surrounding the reef area was the core area (Zone B). The four corner areas of the experimental pond were used as the edge area (Zone A).

Video recording was performed with a high-resolution camera (Logitech, C920 HD Pro Webcam, Qingdao, China) located directly above the experimental pond; the camera was connected to a computer and stored in real-time for monitoring the videos. Video capture was performed using LoliTrack v5 Webcam Software (Loligo^®^ Systems, Copenhagen, Denmark).

Ten fish were randomly selected from the temporary pond and placed in the experimental pond for behavioural monitoring experiments. To avoid adaptive reactions of the experimental fish to the tank environment and reef model, a new batch of 10 fish was replaced after each experiment. To prevent chemical signals from remaining in the water, all the water was replaced after each experiment. Whenever it was necessary to handle animals, such as during transfers between treatments, it was performed carefully with the help of a hand net. All videos were analysed by one observer (Zhanlong Chen), ensuring that the criteria for defining the behaviour of the *S. latus* specimens were the same throughout the analysis process. In all experiments, individual behaviour was indirectly analysed without interference from observers, and videos were recorded through visual webcams.

### 2.4. Daytime Experiment

Our study had 13 groups of experimental setups, with each group consisting of three replicates. Each group included 10 fish per replicate, and a total of 390 juvenile *S. latus* individuals were needed. Among the groups, 12 were set up with one of the 12 artificial reefs described in Section 2.2, and the 13th group was the control group without reefs. To reflect the strain and adaptability of the fish to the natural environment after release, filming began immediately after the fish were placed in the experimental pond, with each group being filmed from 8:00 to 18:00 for a duration of 10 h. HOBO (HOBO MX2022, Onset Computer Corporation, Phoenix, AZ, USA) was used to monitor the light intensity in real-time throughout the experimental process.

### 2.5. Nighttime Experiment

Based on the daytime experiment, artificial reefs with 4 different opening sizes were selected from three reef opening shapes: diamond, circular, and square. They were circular 3-cm reefs, square 6-cm reefs, diamond 9-cm reefs, and circular 12-cm reefs. We set up five groups, each with 3 replicates, with 10 fish in each group and a total of 150 juvenile *S. latus*. Among the five groups, four had one of the different types of reefs described above, and one group was the control group without fish reefs.

The video was started by placing the fish in the experimental pond, and each group was filmed from 20:00 to 6:00, with a duration of 10 h. Filming at night was performed using infrared light. The nighttime light intensity was 0 lx, so no HOBO device was employed. To prevent interference from external factors affecting the light intensity at night, the periphery of the test pond was completely covered with a shading cloth.

### 2.6. Evaluated Parameters and Statistical Analysis

The average behavioural parameters of 5 fish in each of the three repeated experimental groups were randomly selected for statistical analysis. We analysed the reef first contact time with *S. latus* during the 10-h shooting period [27] and the average distribution rate. First contact time was quantified as the time needed for fish to first approach a novel object, explained by the fish’s fear of new objects [28]. In this case, the artificial reef average distribution rate (ADR) was calculated from the average value in three regions to indicate the habitat preferences of *S. latus*, as follows:$$\mathrm{ADR}\;(\%)=\frac{1}{mn}\sum_{i=1}^{m}n_{i}\times 100\%$$
where *n_i_* is the distribution of experimental fish in a certain area during the *i*th observation, *m* is the number of observations, and *n* is the total number of experimental fish.

We used Lolitrack 5 to analyse the motion indicators of *S. latus*, including the average speed (cm/s), average turning rate (deg/s), average distance moved (cm), percentage of active time, nearest neighbour distance (cm), and average interindividual distance (cm) of the fish swimming under different conditions, to determine the swimming behaviour and group activity characteristics of *S. latus*. The average speed, average distance moved, average turning rate, and percentage of the active time were used to illustrate *S. latus* activity, and the nearest neighbour distance and average interindividual distance indicators illustrated the clustering characteristics of *S. latus*.

All parameters are expressed as the mean ± standard deviation (mean ± SD), and *p* < 0.05 was considered to indicate a significant difference. All the statistical analyses were performed using R version 4.3.1. To measure the differences in first contact time among the three reef shapes and four opening ratios, two-way ANOVA was performed, before the ANOVA the Shapiro–Wilk test for normality was performed. Pearson’s correlation analysis was used to analyse the relationship between first contact time and the distribution rate in the reef area. To visualize the relationships between the average distribution rate and the opening shape and opening size, nonmetric multidimensional scaling (NMDS) analysis was performed based on Bray–Curtis distances by using the “metaMDS” function of the “vegan” package [29]. To determine significant differences in ADR between the opening shape and opening ratio, ANOSIM and Adonis tests were conducted using the “anosim” and “adonis” functions, respectively, of the “vegan” package. Then, Pearson correlations were performed between the indicators of swimming behaviour and the reef opening ratio to evaluate consistency across contexts (between behaviour and reef type). The F test was used to analyse the difference between daytime and nighttime behaviour indicators. The differences in reef type (opening ratio and opening shape) and light intensity and differences in the behavioural indicators were calculated, and Spearman’s correlation analysis and a multiple linear regression model were combined with variance decomposition to analyse the contributions of the main reef type and light intensity to the behavioural indicators.

All parameters are expressed as the mean ± standard deviation (mean ± SD), and *p* < 0.05 was considered to indicate a significant difference. All the statistical analyses were performed using R version 4.3.1. To measure the differences in first contact time among the three reef shapes and four opening ratios, two-way ANOVA was performed, before the ANOVA the Shapiro–Wilk test for normality was performed. Pearson’s correlation analysis was used to analyse the relationship between first contact time and the distribution rate in the reef area. To visualize the relationships between the average distribution rate and the opening shape and opening size, nonmetric multidimensional scaling (NMDS) analysis was performed based on Bray–Curtis distances by using the “metaMDS” function of the “vegan” package [29]. To determine significant differences in ADR between the opening shape and opening ratio, ANOSIM and Adonis tests were conducted using the “anosim” and “adonis” functions, respectively, of the “vegan” package. Then, Pearson correlations were performed between the indicators of swimming behaviour and the reef opening ratio to evaluate consistency across contexts (between behaviour and reef type). The F test was used to analyse the difference between daytime and nighttime behaviour indicators. The differences in reef type (opening ratio and opening shape) and light intensity and differences in the behavioural indicators were calculated, and Spearman’s correlation analysis and a multiple linear regression model were combined with variance decomposition to analyse the contributions of the main reef type and light intensity to the behavioural indicators.

## 3. Results

### 3.1. First Contact Time

According to the first contact time between the experimental fish and the reef, different reef structures and day and night conditions affected the neophobic and adventitious behaviour of the experimental fish (Figure 3). The Shapiro–Wilk test for normality was performed before the ANOVA (Figure 4). The *p*-values were all greater than 0.05, which meets the homogeneity test. In the diamond-shaped reef treatment group, the fish had the strongest adventurous behaviour, with an average time of 245.88 s for first contact with the fish reef. Then, in the circular reef treatment group, the response time was 513.25 s, and the slowest response time was found in the square reef treatment group, which was 783 s. Pearson’s correlation analysis revealed that the fear of the novelty behaviour of fish was negatively correlated (*p* < 0.005) with the distribution rate in the artificial reef area (Figure 5). The shorter the initial exploration time was, the greater the average distribution rate in the artificial reef area. In addition, the first contact time of *S. latus* at night was shorter than that during the day under the corresponding reef shape (Figure 3c), indicating that light conditions constitute one of the external conditions affecting the first contact time of *S. latus*.

### 3.2. Habitat Preferences

There were differences in the habitat preferences of *S. latus* under the different habitat conditions. When artificial reefs were added, the average distribution rate in Zone C significantly increased. In the control group, the average distribution rate of *S. latus* in each region was 33.83% in Zone A, 60.83% in Zone B, and 5.33% in Zone C, showing the following trend: Zone B > Zone A > Zone C. The preference of the experimental fish was distributed in the middle and edge areas of the experimental pond, while the average distribution rate in Zone C was significantly lower. After adding artificial reefs, the average distribution rate of *S. latus* in various regions was 44.10% in Zone B, 39.82% in Zone A, and 16.08% in Zone C. The artificial reefs had a significant trapping effect on *S. latus* (Figure 6). Among the three types of reef structures, the average distribution rate in Zone C of the diamond reef was greater than that of the corresponding opening ratios of the circular and square reefs, indicating that among the reefs with the three types of opening shapes, the diamond reef exerted a greater impact on *S. latus* and a greater trapping effect. When the opening ratio was 1.5, the average distribution rate of *S. latus* in Zone C was the highest, at 25.44%, indicating that the opening ratio of the reef was 1.5 times greater than that of the fish body and that *S. latus* exhibited a greater reef preference. The treatment groups exhibited opening ratios of 2.0 and 1.0. The average distribution rates in Zone C were 19.11% and 12.06%, respectively, and the lowest value was 7.72% at an opening ratio of 0.5.

The presence or absence of environmental enrichment in the habitat significantly affected the average distribution rate of *S. latus*, further affecting its habitat preferences. An NMDS plot was created based on the binary Bray–Curtis similarity index displaying the average distribution rate between the opening ratio and opening shape.

The fitting results for values of nonsimilarity of observation and sorting distance (Figure 7) showed that no points appeared far from the line segment, indicating that the data can be analysed using NMDS analysis. The stress in the regression line was 0.028, which is much less than 0.2, further indicating that NMDS is suitable for analysing the differences in average distribution rates under different reef structures and sizes. The average distribution rate relationships were visualized via NMDS analysis (Figure 8). These ordination graphs revealed different preference characteristics for the different control groups, opening ratios and opening shapes, as confirmed by permutational multivariate analysis of variance (Adonis: opening size: *R*^2^ = 0.36, *p* = 0.009; opening shape: *R*^2^ = 0.12, *p* = 0.015) and similarity analysis (ANOSIM: opening size: *R* = 0.28, *p* = 0.008; opening shape: *R* = 0.06, *p* = 0.021).

There was a significant difference in the average distribution rate between the group without artificial reefs and the group with artificial reefs. The different opening ratios (Adonis, *R*^2^ = 0.36) had greater explanatory power for the difference in the *S. latus* distribution than did the opening shape (Adonis, *R*^2^ = 0.12), indicating that the opening ratio of the reefs was more likely to affect the habitat preference characteristics of *S. latus*.

### 3.3. Swimming Behaviour

According to follow-up bivariate regressions, in artificial reefs with different opening shapes, the average speed, average distance moved, average interindividual distance, nearest neighbour distance and active percentage of time decreased gradually with increasing opening ratio. Nonetheless, the average turning rate movement increased gradually. In diamond reefs only, significant and negative linear relationships were found between the avg. speed (*p* < 0.05), avg. distance moved (*p* < 0.05) and avg. inter-individual distance (*p* < 0.05) and opening ratio (Figure 9a–c).

There were differences between the daytime and nighttime movement behaviours of *S. latus* (Figure 10). In terms of movement indicators, the average turning rate and average distance moved significantly differed between day and night for the different reef structures. In an environment without artificial reefs, the average speed and distance of *S. latus* at night were lower than those during the day, indicating that *S. latus* has a strong activity ability during the day. After the addition of artificial reefs, the average speed and distance at night generally increased compared to those during the day, indicating that *S. latus* has a strong nocturnal activity ability. The nearest neighbour distance and average interindividual distance decreased compared to those in the daytime, indicating that, on the one hand, the addition of artificial reefs changed the movement characteristics of *S. latus*, and at the same time, light intensity affected the movement behaviour. Overall, when there were no enrichment habitat conditions, the juvenile *S. latus* tended to move more during the day, with poor clustering. When there was enrichment in habitat conditions, *S. latus* preferred to move in groups at night.

### 3.4. Driving Factors of Behaviour

Spearman’s correlation analyses were conducted to explore the possible influence of the opening ratio and light intensity on the reef first contact time, the distribution rate and the 6 movement indicators used (Figure 11). The opening ratio was negatively correlated with the average distribution rate and average neighbour individual distance of *S. latus* in Zone A, while it was highly significantly positively correlated with the average distribution rate in Zone C, indicating that the opening ratio significantly affected the reef approach and clustering of *S. latus*. Light intensity was negatively correlated with the average distance moved and average speed, while it was significantly positively correlated with the first contact time, indicating that light intensity affected the exploratory and activity ability of *S. latus*.

## 4. Discussion

### 4.1. Importance of Artificial Reef Habitats in the Natural Environment

Artificial reefs can provide necessary and safe habitats for reef fishes and play a crucial role in their adaptability and survival [30]. The results of this study indicate that the addition of artificial reefs increased the average distribution rate of *Sparus latus* in the middle zone of the experimental pond. During the breeding process, *Sparus aurata* often swim around the net wall in empty breeding cages and evidently do not frequently use the centre of the breeding pond [31]. The addition of artificial reefs provides physical complexity and shelter space for the habitat environment, promotes spatial exploration, and alters the distribution characteristics of fish [32]. Taking the reef preference characteristics of *S. latus* as a welfare indicator for habitat enrichment, the tendency of *S. latus* to use reefs increases and greater survival benefits are obtained through the process of behavioural adaptation. Consistent with this, studies on *Gadus morhua* have shown that cod raised in an enriched structure exhibit reduced swimming activity, more environmentally dependent variations in the fish population, and an increase in their activity time in the enriched structure [33]. *Acanthopagrus schlegelii*, as a reef fish, prefers to hide around reefs or obstacles to avoid rapids and save energy [34]. Some studies have shown that when artificial reefs are placed, their attractive effect is more significant than that of the control group [35]. At the same time, studies on fish such as *Pagrus major*, *Plectorhinchus cinctus*, and *Hexagrammos otakii* have shown that artificial reefs have similar attractive effects [30,36,37].

The movement trajectory of fish can to some extent reflect whether they are suitable for the environmental conditions they are in, and swimming indicators such as the speed of fish movement in water are important indicators of fish behaviour [38]. The results of this study indicate that after being placed on artificial reefs, the average speed and average distance moved by *S. latus* decreased. This indicates that the creation of artificial habitats is crucial for *S. latus*, as they have a certain attraction effect, and thus, the fish are more willing to spend time wandering around and inside the reef, which makes the overall two movement indicators lower in the *S. latus* group than in the control group. This finding is consistent with existing studies showing that after the habitat becomes complex, the swimming activity of *Salmo trutta* in open spaces decreases, and the movement of released brown trout *S. trutta* in natural rivers decreases compared to that in typical barren environments in hatcheries [39]. Compared to an uncovered environment, *Salmo salar* prefers to remain in artificial habitats and reduce its motor behaviour and metabolic rate [40]. In addition, the performance of approaching reefs varies with the opening size, and it was most significant when the opening size was 1.5 times the height of the fish body. *Acanthopagrus schlegelii*, which is also a reef-dwelling fish, prefers reefs with a slight difference between the side length of the opening and the body height, mainly due to the shielding effect of reefs. It is considered that a larger surface area of reefs could provide a better shielding effect of reefs [32].

### 4.2. Role of Shoaling Behaviour in Habitat Enrichment

Shoaling is an important behaviour of fish and plays a crucial role in the survival of most fish. However, the shoaling behaviour of fish is not stable and is influenced by various environmental factors, such as density, temperature, light intensity, and habitat. The population size of juvenile *Hypophthalmichthys nobilis* has a significant impact on the nearest neighbour distance and spatial distribution, and the cohesion of the population increases with increasing population size [41], indicating that population density affects the clustering behaviour of the bighead carp *H. nobilis*. Our study showed that the number of experimental fish in each group was the same, so we failed to reflect the effect of density on the aggregation behaviour of *S. latus*. Individual spacing is a key behavioural indicator representing the behaviour of fish clusters. Based on distance data between individuals, it was found that individual size differences do not affect the cohesion of juvenile populations of *Parabrami spekinensis* and *Spinibarbus sinensis* [42]. The light intensity is an important factor that affects the shoaling behaviour of *Sebastes schlegelii*. Studies have shown that more than 90% of *S. schlegelii* exhibit the highest group cohesion under 0 lx light conditions [43]. We found that *S. latus* notably clustered at night and weakly during the day.

In addition, we found that the clustering behaviour of *S. latus* was influenced by habitat enrichment. When there is no enriched structure in the habitat, *S. latus* tends to be active during the day and does not cluster, while *S. latus* tends to engage in cluster activities at night. We use the nearest neighbour distance to measure the cohesion of fish clustering behaviour. The clustering behaviour of the brown trout *S. trutta* and *Theragra chalcograma* becomes looser with increasing temperature, and the distance between neighbouring fish and individuals has recently increased, resulting in a decrease in cluster cohesion [44,45]. When the water body rapidly warmed, there was no effect on the average interindividual distance or nearest neighbour distance of *Carassius auratus gibelio*, indicating that water temperature had no significant effect on population cohesion or coordination. However, as the complexity of the ecological situation increases, the swimming speed, synchronicity, and cluster cohesion of individuals show a downward trend [46]. This is different from the results of this study, and we found that the nearest neighbour distance of *S. latus* in the experiment never exceeded the two body lengths, indicating that the addition of artificial reefs does not affect the cohesion of the population, which is consistent with the findings for the reef taxis of *S. latus*. When faced with predators, the clustering behaviour of *Phoxinus phoxinus* in simple structured habitats is significantly greater than that without predators, and this phenomenon was not found in complex habitats, indicating that the clustering behaviour is not as obvious when there is sufficient physical structure in the habitat [47]. Compared with our results, this is mainly due to the addition of a third external condition, namely, predator, to the environmental factors. Therefore, in more complex environments, the formation conditions of fish clustering behaviour become more complex.

### 4.3. Impact of Habitat Enrichment on First Contact Time

Introducing new objects into a habitat may lead to a negative psychological state in certain fish, known as first contact time [31]. Fish may experience reduced activity or exploratory behaviour due to fear of the new environment, or territorial behaviour may be enhanced due to the scarcity of structures introduced into the new environment [48,49]. The independent biological functions involved in animal responses to novel things, such as exploration, anxiety, and fear of novelty, are influenced by differences in animal-rearing environments [50]. Generally, due to the oppression of natural environments such as predators in the wild, wild fish have lower courage, lower activity levels, and a greater fear of novelty [28]. In the aquaculture environment, fish are subjected to the same external environmental pressure, but the degree of enrichment will affect the level of fish first contact time. Fish with higher environmental enrichment levels exhibit greater courage, higher activity levels, and lower levels of neophobia [51]. This is mainly achieved by using habitat enrichment technology to habituate animals to novel environments, thereby increasing their courage and activity while reducing their fear of new environments [52]. We did not observe an effect of artificial reef structures on the neophobic behaviour of *S. latus*. The fish showed similar courage and neophobic behaviour in the three reef structure environments. On the one hand, because the reef structure and quantity were not different, they were all cubic structures. On the other hand, the fact that the fish were all from the same environmental pressure breeding conditions indicates that the same habitat enrichment structure and external pressure will not affect neophobic behaviour.

The most direct explanation for the differences in fish behaviour such as neophobia is genetic factors rather than environmental factors, which only influence such behaviour in a fine-tuning manner [53]. Our study showed that day/night differences are important factors in altering the first contact time of *S. latus*. The results showed that the first contact time of *S. latus* at night was relatively low, indicating that a low light intensity increases the exploration behaviour of fish, which may be related to the resistance of *S. latus* to light. Under genetic control, changes in light intensity affect the fear of new objects [52]. However, animals’ reactions to new objects are interrelated, as anxiety or fear of novelty behaviours are accompanied by exploratory or novel preference behaviours [54,55]. An exploratory nature is commonly encountered in fish, as it allows them to obtain information on predators and food resources, assist them in finding suitable habitats, and reduce the risk of being preyed upon [56]. The exploration behaviour of fish often changes due to individual factors and their living environment (life history experience) [57]. Environmental enrichment reduces the duration of captive *G. morhua* to recover from exposure to stressors and increases their exploratory behaviour towards new objects [58]. Enrichment could reduce fear, increase exploration of unknown spaces, and reduce the stress responses to environmental stressors [59,60]. Therefore, the effectiveness of habitat enrichment for aquaculture depends on the behaviour of the species involved.

### 4.4. Driving Factors of Habitat Enrichment for Sparus latus Behaviour

Habitat enrichment technology has different effects on species of cultured organisms, and this difference depends not only on the life stage of fish, but also on enrichment structure and aquaculture characteristics [31]. The activity and survival status of *S. asotus* is influenced not only by differences in shelter structures within the breeding environment but also by the breeding density, experimental time, and breeding environment, reflecting the ecology of the species [61]. In this study, light intensity was a significant negative factor affecting the average speed and average distance moved by yellowfin bream, while a study of *S. schlegelii*, which is also a reef-dwelling fish, showed that its average speed increased with increasing light intensity [30]. This may be due to the influence of the fish’s own circadian rhythm personality. Different behavioural patterns are observed among different species, with *Rutilus rutilus* exhibiting higher swimming levels in the early morning and evening, while swimming levels decrease at night [62]. However, *Tinca tinca* and *Corydoras amphibelus* exhibit increased nocturnal activity [63]. The most significant factor affecting the habitat selectivity of *S. latus* is the opening ratio, as the larger the opening ratio is, the stronger the tendency to use reefs. This study revealed that juveniles prefer reefs with slight differences in the aperture length and body height. It is believed that the vertical plate of the reef may serve as an obstacle, and a larger solid surface area can provide a better shielding effect [37]. In addition, there was a significant negative correlation (*p* < 0.01) between the opening ratio and the average neighbour fish distance, indicating that the larger the opening is, the weaker the clustering behaviour of *S. latus*.

These results reveal a complex interplay between different types of environmental heterogeneity and subsequent behavioural responses. We found that without habitat enrichment, *S. latus* prefers to remain active during the day without clustering, while with habitat enrichment, *S. latus* prefers to remain active at night with cluster-like activity. Compared to the light intensity, artificial reefs are more likely to be positively correlated with the aggregation behaviour of *S. marmoratus* [64]. This further supports that the behavioural responses of fish were explored by testing fish responses to small-scale artificial reef structures in laboratory experiments [65].

## 5. Conclusions

In summary, the behavioural strategies and driving factors of the response of juvenile *Sparus latus* to enrichment structure were explored by testing fish responses to small-scale artificial reef structures in laboratory experiments. We found that when comparing day- and nighttime behaviours, habitat enrichment altered the habitat preferences and movement indicators of *S. latus*. The fish shifted from being more active during the daytime to being more active at night, while their clustering behaviour also increased at night. Moreover, the first contact time was lower at night. Spearman’s correlation analysis revealed that the exploratory ability of juvenile fish is influenced by the opening shape and light intensity, while reef taxis and clustering are influenced by the opening ratio. In addition, the activity ability of juvenile fish was mainly affected by the light intensity. Overall, the impact of the enrichment structure on the different personalities and behavioural abilities of fish was complex. Therefore, it is necessary to carefully adjust the habitat enrichment solution based on the actual scenario in which the sea fish are released to adapt to biological needs and cultivate farmed fish in conditions that are closer to the natural environment and therefore improve the environmental adaptability and survival rate of juvenile fish after release.

## Figures and Tables

**Figure 1 biology-13-00364-f001:**
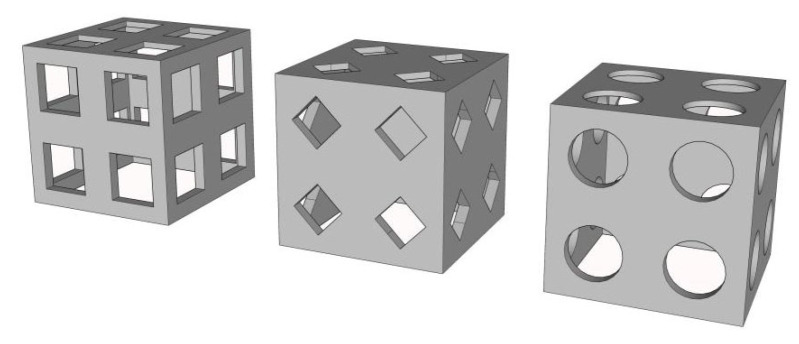
Diagram of the artificial reef model.

**Figure 2 biology-13-00364-f002:**
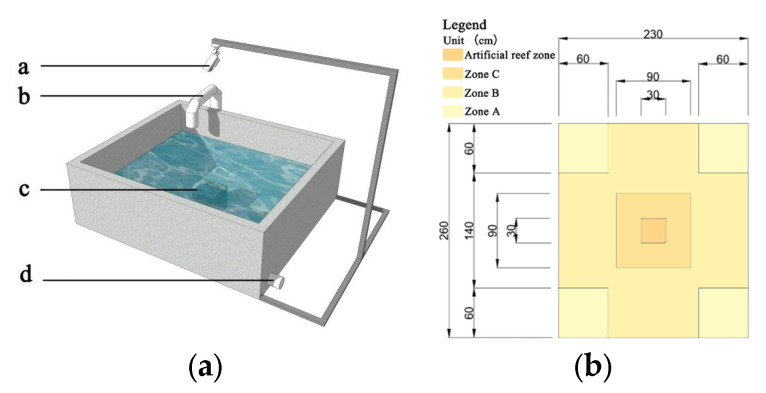
Diagram of the experimental pond (**a**) and diagram of bottom area division (**b**). In (**a**): a. Camera; b. inlet; c. artificial reef; d. outlet.

**Figure 3 biology-13-00364-f003:**
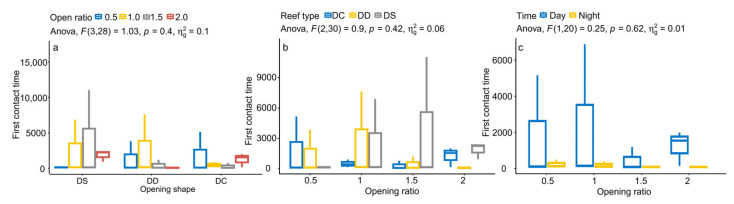
Effects of the opening shape, opening ratio and diurnal difference on reef first contact time of *S. latus.* (**a**) Effect of opening shape on first contact time; (**b**) Effect of opening ratio on first contact time; (**c**) Effect of diurnal difference on first contact time.

**Figure 4 biology-13-00364-f004:**
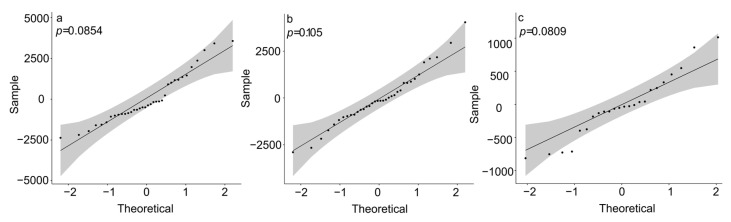
Shapiro–Wilk test for the opening ratio (**a**), opening shape (**b**) and time (**c**).

**Figure 5 biology-13-00364-f005:**
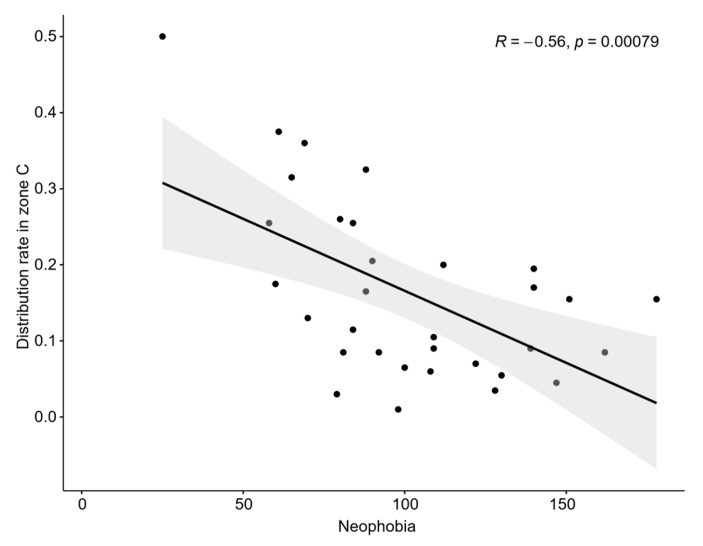
Relationship between first contact time and the distribution rate in artificial reef areas.

**Figure 6 biology-13-00364-f006:**
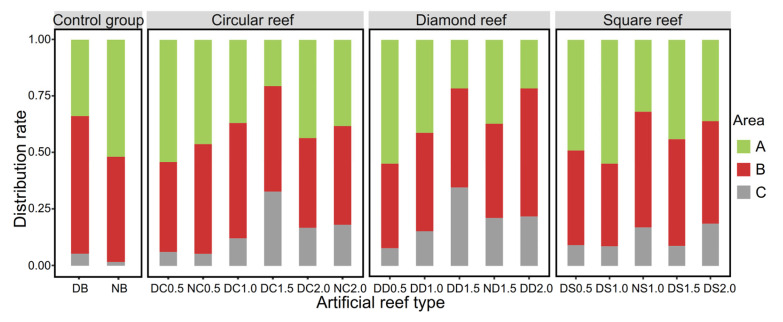
Average distribution rate of *S. latus* for the different types of artificial reefs. In terms of the first letter of the horizontal coordinate, D denotes day, N denotes night, and in regard to the second letter, B, C, D, and S denote the control group, circular reef, diamond-shaped reef, and square reef, respectively. The values of 0.5~2.0 denote the open ratio.

**Figure 7 biology-13-00364-f007:**
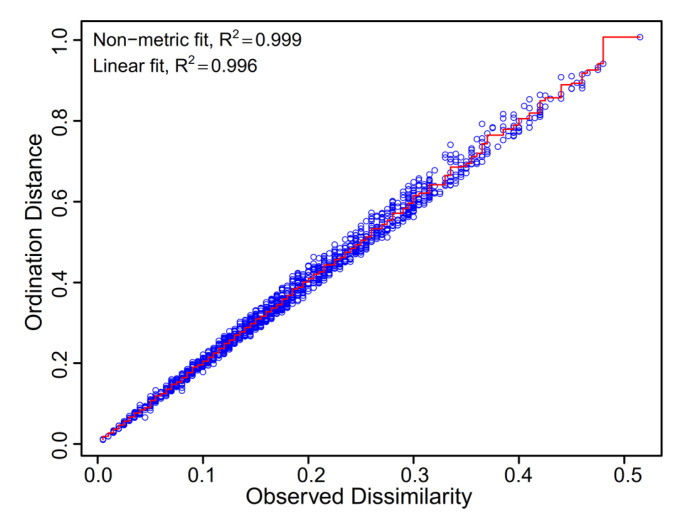
Shepard diagram of the regression relationship between the observed dissimilarity and ranking distance.

**Figure 8 biology-13-00364-f008:**
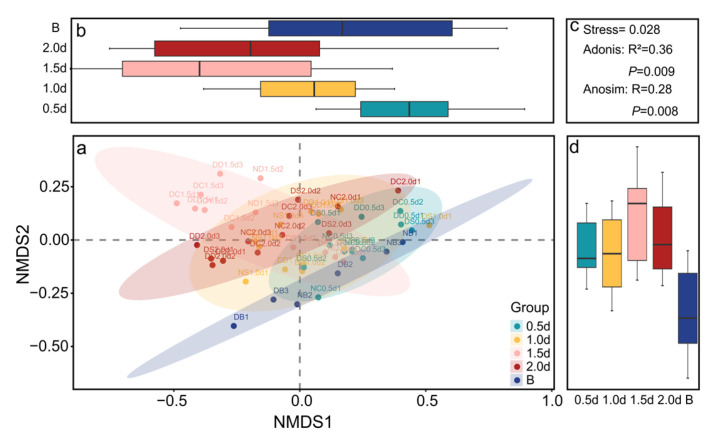
Nonmetric multidimensional scaling (NMDS) plot based on the binary Bray–Curtis similarity index displaying the average distribution rate between the opening ratio (**a**) and opening shape (**e**). (**b**,**d**) Grouped box diagrams of the NMDS1 and NMDS2 values of the opening ratio indicators. (**c**,**g**) Anosim and Adonis Non-parametric test coefficients. (**f**,**h**) Grouped box diagrams of the NMDS1 and NMDS2 values of the opening shape indicators.

**Figure 9 biology-13-00364-f009:**
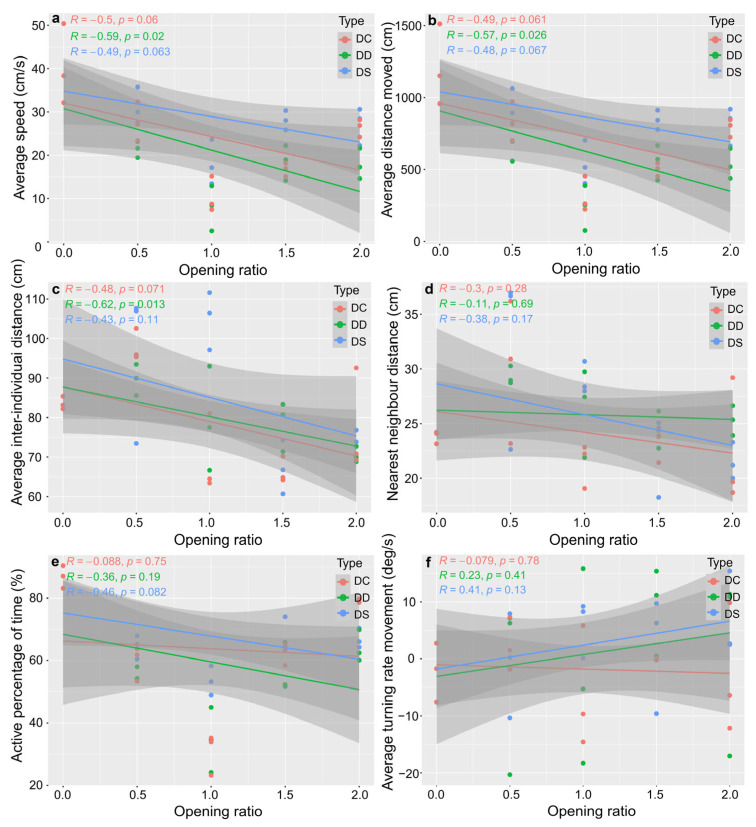
Correlation analysis between the 6 indicators of swimming behaviour and reef shape and the opening ratio of *S. latus*. The solid line denotes a significant linear behavioural response between artificial reef types, and the grey shape denotes the 95% confidence interval. The *p* value and R (Pearson) are shown for linear regression. (**a**) Average speed correlation; (**b**) Average distance moved correlation; (**c**) Average inter-individual distance correlation; (**d**) Nearest neighbour distance correlation; (**e**) Active percentage of time correlation; (**f**) Average turning rate movement correlation.

**Figure 10 biology-13-00364-f010:**
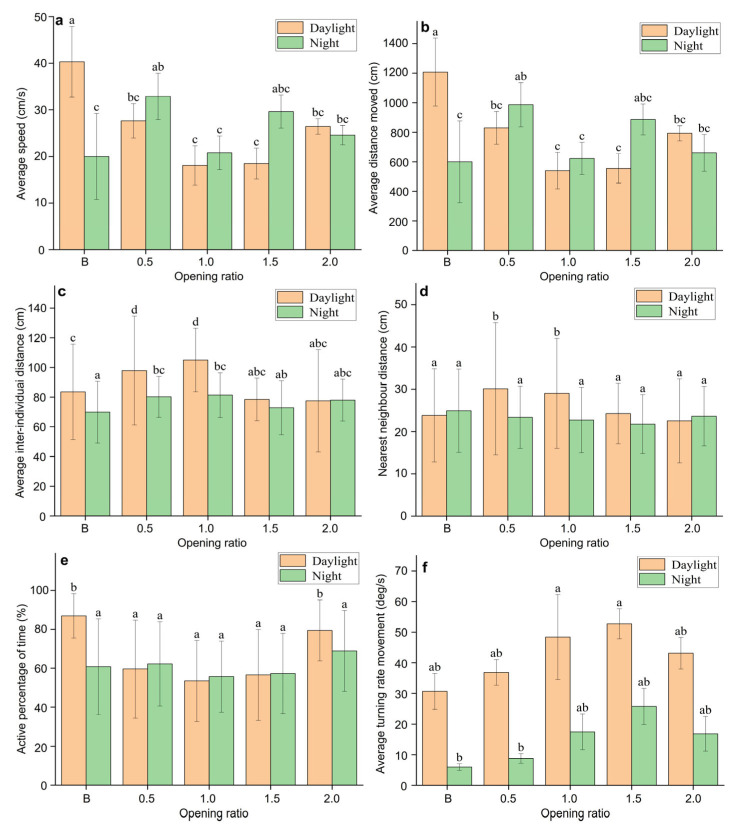
Analysis of the diurnal differences in the 6 indicators of swimming behaviour of *S. latus.* The horizontal axis indicates the opening ratio, where B is the control group. (**a**) Diurnal differences in average speed; (**b**) Diurnal differences in average distance moved; (**c**) Diurnal differences in average inter-individual distance; (**d**) Diurnal differences in nearest neighbour distance; (**e**) Diurnal differences in active percentage of time; (**f**) Diurnal differences in average turning rate movement. Different superscripts on the bars denotes statistically significant differences (*p* < 0.05).

**Figure 11 biology-13-00364-f011:**
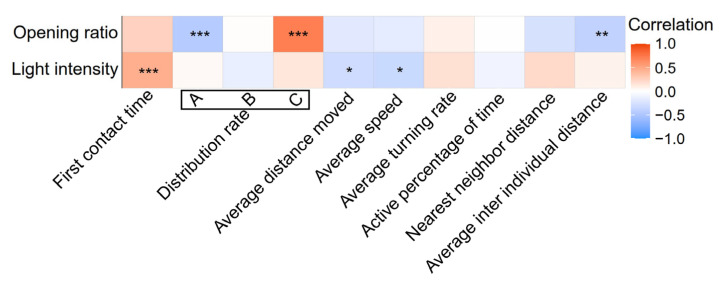
Heatmap of the Spearman’s correlation coefficients of the opening ratio and light intensity parameters and behavioural characteristics of *S. latus* (colours denote Spearman’s correlation coefficients), with the symbol indicating the significance of the linear regression (* 0.01 < *p* < 0.05, ** 0.001 < *p* < 0.01 and *** *p* < 0.001).

## Data Availability

Data are available upon request to the corresponding author.

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
