# Peer review of "Habitat Enrichment Causes Changes in Fish Behavioural Characteristics: A Case Study of Sparus latus"

_biology, 2024, doi:10.3390/biology13060364_

Round 1

Reviewer 1 Report

Comments and Suggestions for Authors

Habitat enrichment changes fish behavioural characteristics: A case study of Sparus latus by Yu Guo et al. – biology-2912430

The study deals with the effects of different types of artificial reefs as habitat enrichment conditions on the habitat preference and the behavioural strategies of juvenile S. latus, as well as the driving factors affecting the behaviours. The paper covers an interesting issue and provides useful information, which would be essential for the design of suitable environments for the growth and survival of fish and therefore for the maintenance and conservation of biodiversity. In my opinion, It requires some improvements before publication. Suggestions or comments to improve the document are listed below.

L77 What is meant by "Paralichthys olivaceus prefers to inhabit sandy, rocky, or rocky areas."? Please rephrase.

L78-79 "The deployment of artificial reefs provides essential habitats for reef fish, including spawning, feeding, nurturing, and shelter." Please rephrase.

L83 development = survival ? It is unclear.

L94-95 "the habitat selection preferences and behavioural strategies" = "the habitat selection and behavioural strategies".

L88-99 This paragraph is important and still needs to be rewritten and improved with the specific objectives of the study being clearly defined and presented.

L113 17 am = 5pm?

L124"The open rate was 0.5 times, 1.0 times, 1.5 times and 2 times" = "The open rate was 0.5, 1.0, 1.5 and 2 times".

L158 “Daytime environmental enrichment experiment”  = “Daytime experiment”

L168 “Nighttime environmental enrichment experiment” = “Nighttime experiment”

L182-183 “"Neophobia was quantified as how much time a fish first took to approach a novel object, explained by the fish's fear of new things." authors should provide a reference.

L184-185 "... within three regions to indicate S. latus preference for habitat preference, and..."  = "...in three regions to indicate the habitat preference of A. lotus, and..."

L196 "... S. latus activity ability,..."  = "...S. latus activity ability,..."

L230 "... Sparus latus..."  = "...S. latus..."

L236 The resolution and clarity of Fig 3. could be improved. Authors should also label Figure 3 as a, b and c, which should also be indicated in the caption (see e.g. Figures 6 and 7).

L275 NMDS = nonmetric multidimensional scaling (NMDS)

L278  nonmetric multidimensional scaling (NMDS) = NMDS

L299 Swimming behaviour characteristics = Swimming behaviours

L307 Standardize the font size of the Y titles in Fig. 7.

L354-356 “The addition of artificial reefs provides physical complexity and shelter space for the habitat environment, promotes spatial exploration, changes the distribution characteristics of fish (increases the use of ponds), and provides more space between fishing nets and fish schools” is confusing. Please rephrase.

L363-364 "Acanthopagrus schlegelii, as a reef fish, prefers to hide around underwater reefs or obstacles to avoid rapids and save energy.", authors should provide a reference.

L383 “clustering behavior = shoaling behavior”, here and elsewhere, please check.

L397-398 "We found that S. latus exhibits significant nocturnal clustering  behaviour and weaker daytime clustering behaviour.", please rephrase.

L342 Fig. 9 can be enlarged.

L466 “complexity of the increase = complexity of rearing”, please check.

l478 "(p<0.001)", this can be removed.

L479-481 "Studies have shown that juveniles tend to prefer reefs with small differences between aperture length and their body height.", please provide references.

L532-662 I encourage authors to consult the MDPI's guide for authors to improve the presentation of the reference list.

Author Response

Thank you very much for your suggestions and comments on our manuscript. These comments have effectively improved our manuscript. We have included almost all of the suggestions, and below, we present our point-by-point responses to the provided comments.

The study deals with the effects of different types of artificial reefs as habitat enrichment conditions on the habitat preference and the behavioural strategies of juvenile S. latus, as well as the driving factors affecting the behaviours. The paper covers an interesting issue and provides useful information, which would be essential for the design of suitable environments for the growth and survival of fish and therefore for the maintenance and conservation of biodiversity. In my opinion, It requires some improvements before publication. Suggestions or comments to improve the document are listed below.

  1. L77 What is meant by "Paralichthys olivaceus prefers to inhabit sandy, rocky, or rocky areas."? Please rephrase.

Response to comments: Thank you for your suggestions. The manuscript has been revised accordingly (L60).

  1. "The deployment of artificial reefs provides essential habitats for reef fish, including spawning, feeding, nurturing, and shelter." Please rephrase.

Response to comments: Thank you for your advice. This sentence has been rephrased in the revised manuscript (L62-63).

  1. L83 development = survival ? It is unclear.

Response to comments: Thank you for your comments. We have revised this statement (L67).

  1. L94-95 "the habitat selection preferences and behavioural strategies" = "the habitat selection and behavioural strategies".

Response to comments: Thank you for your comments. We have made changes according to your suggestion in the revised manuscript (L104).

  1. L88-99 This paragraph is important and still needs to be rewritten and improved with the specific objectives of the study being clearly defined and presented.

Response to comments: Thank you for your comments. According to your suggestion, we have improved this paragraph in the revised manuscript (L88-109).

  1. L113 17 am = 5pm?

Response to comments: Thank you for your careful review. This term has been modified in the revised manuscript (L122-123).

  1. "The open rate was 0.5 times, 1.0 times, 1.5 times and 2 times" = "The open rate was 0.5, 1.0, 1.5 and 2 times".

Response to comments: Thank you for your comments. This term has been modified in the revised manuscript (L133).

  1. L158 “Daytime environmental enrichment experiment”  = “Daytime experiment”.

Response to comments: Thank you for your comments. This term has been modified in the revised manuscript (L167).

  1. L168 “Nighttime environmental enrichment experiment” = “Nighttime experiment”

Response to comments: Thank you for your careful review. This term has been modified in the revised manuscript (L177).

  1. L182-183 “"Neophobia was quantified as how much time a fish first took to approach a novel object, explained by the fish's fear of new things." authors should provide a reference.

Response to comments: Thank you for your comments. We have supplemented the references in the revised manuscript (L194).

  1. L184-185 "... within three regions to indicate S. latus preference for habitat preference, and..."  = "...in three regions to indicate the habitat preference of A. lotus, and..."

Response to comments: Thank you for your comments. This term has been modified in the revised manuscript (L195-196).

  1. L196 "... S. latus activity ability,..."  = "... latusactivity ability,...".

Response to comments: Thank you for your comments. This term has been modified in the revised manuscript (L206).

  1. L230 "... Sparus latus..."  = "... latus..."

Response to comments: Thank you for your careful review. This term has been modified in the revised manuscript (L240).

  1. L236 The resolution and clarity of Fig 3. could be improved. Authors should also label Figure 3 as a, b and c, which should also be indicated in the caption (see e.g. Figures 6 and 7).

Response to comments: Thank you for your suggestions. In the revised manuscript (L244-245), we have modified Figure 3 and added panels a, b and c.

  1. L275 NMDS = nonmetric multidimensional scaling (NMDS)

Response to comments: Thank you for your comments. It has been explained in L215 of section 2.6. So this term has been modified in the revised manuscript (L281).

  1. L278  nonmetric multidimensional scaling (NMDS) = NMDS.

Response to comments: Thank you for your comments. This term has been modified in the revised manuscript (L282).

  1. L299 Swimming behaviour characteristics = Swimming behaviours

Response to comments: Thank you for your careful review. This term has been modified in the revised manuscript (L304).

  1. L307 Standardize the font size of the Y titles in Fig. 7

Response to comments: Thank you for your careful assessment. This term has been modified in the revised manuscript (L312).

  1. L354-356 “The addition of artificial reefs provides physical complexity and shelter space for the habitat environment, promotes spatial exploration, changes the distribution characteristics of fish (increases the use of ponds), and provides more space between fishing nets and fish schools” is confusing. Please rephrase.

Response to comments: Thank you for your comments. We have changed this part to " The addition of artificial reefs provides physical complexity and shelter space for the habitat environment, promotes spatial exploration, and alters the distribution characteristics of fish" (L357-358).

  1. L363-364 "Acanthopagrus schlegelii, as a reef fish, prefers to hide around underwater reefs or obstacles to avoid rapids and save energy.", authors should provide a reference.

Response to comments: Thank you for your comments. We have supplemented the references in the revised manuscript (L365-366).

  1. L383 “clustering behavior = shoaling behavior”, here and elsewhere, please check.

Response to comments: Thank you for your careful review. We have completely replaced this part in the revised manuscript (L391-392, L402-403).

  1. L397-398 "We found that S. latus exhibits significant nocturnal clustering  behaviour and weaker daytime clustering behaviour.", please rephrase.

Response to comments: Thank you for your suggestions. In the revised manuscript (L405), we have rephrased this sentence as: “We found that S. latus notably clustered at night and weakly clustered during the day”.

  1. L342 Fig. 9 can be enlarged.

Response to comments: Thank you for your comments. This term has been modified in the revised manuscript (L345).

  1. L466 “complexity of the increase = complexity of rearing”, please check

Response to comments: Thank you for your careful evaluation. This term has been modified in the revised manuscript (L472).

  1. L478 "(p<0.001)", this can be removed.

Response to comments: Thank you for your comments. This term has been removed from the revised manuscript (L483-485).

  1. L479-481 "Studies have shown that juveniles tend to prefer reefs with small differences between aperture length and their body height.", please provide references.

Response to comments: Thank you for your comments. This conclusion was obtained through our study, and it is summarized again in the manuscript (L483-485) to avoid confusion.

  1. L532-662 I encourage authors to consult the MDPI's guide for authors to improve the presentation of the reference list.

Response to comments: Thank you for your useful suggestions. We have improved all the references in the revised manuscript.

Reviewer 2 Report

Comments and Suggestions for Authors

line 21: The first abstract sentence is very long and difficult to get specific information from. I would change this to something like: "In order to gain a better understanding of Sparus latus habitat preference and behavioural ecology, we performed an experiment using box-shaped reefs as habitat enrichment materials, allowing us to determine the behavioural strategies and drivers in response to different enrichment structures."

line 24-25: no significant difference between what?

line 25: to determine that a phobia is specifically for something neo/new, then a more complex experimental design would be needed, e.g. a range of different new things could be provided, and it should be checked if the phobia was for all of them. If only one new thing is provided, then the phobia may be for that specific thing, and if another different thing was newly provided, then maybe there would be no phobia. This certainly cannot be know if only one kind of new thing was provided as part of the experimental design. Overall, it may be better to use a different term here instead of "neophobia". The term "reef taxis" is used a number of times in the manuscript and I would suggest only using this term, as its accurate for what was actually measured in these experiments, more so than the term neophobia.

line 27: what is exactly the control group? Is it the experimental treatment where no kind of enrichment structure was provided? Please specify here.

line 32: shape is normally a categorical variable, so I would think a correlation analysis (requiring two continuous variables) is not correct here.

line 81: please provide the scientific name for Korean rockfish here.

line 96: please clarify the term "abundance pore structure"

line 119: what colour was the acrylic?

line 123: the term "diameter" is only applicable to the circular shape. Possibly it would have been better to calculate the area of the opening and standardise among shapes using that area, rather than the "opening ratio" used here (e.g. the reefs with square openings have the greatest area of opening overall, more than the other treatments, even those with the same "opening ratio").

line 123: so the reefs with opening diameter of 3cm had openings basically too small for the fish (height about 6cm) to use?

line 171: Is it meant here that only these specific shapes/sizes were used? Why not use all the combinations of shapes and sizes as was done in the daytime experiment?

line 172: it states each experimental set up had three replicates. Please can you confirm here or somewhere else in the methods whether the average parameters of all fish in each replicate were used for the statistical analyses, or whether data from each individual fish within the replicate were used for the analyses. 

line 176: I guess the filming at night was done using infra red light, if so please state that here. And also, it would be important that none of the light is near-infrared, which the fish may be sensitive to. If it cannot be confirmed from past research that the fish are not affected by any near-infrared light of such a camera, then this is something that would need to be tested as part of the current study.

line 183: but the filming began straight away after the fish were put in the experimental enclosure (line 164) so basically everything in the enclosure, and even the enclosure itself, was a novel object.

line 202: The assumption of homogeneity of variances should be tested before ANOVA like this. For testing this assumption, please use Cochran's test or Levene's test. If the assumption is not met, then the data may need to be transformed and it may influence the results and conclusions.

line 218: does contact time mean when a fish physically touches the acrylic reef? I would have thought that some measurement would have been taken about the fish actually going inside the reef. Why was this not measured?

line 219: I do not understand why its stated here that different reef structures had an effect, but then the next sentence states "under different reef opening ratios and shapes, there was no significant difference in the time when the experimental fish first came into contact with the reef". These two statements seem to contradict.

line 222: please provide all the details for the ANOVA statistical output (e.g. sums of squares, F values, specific P values). In particular, seeing as this was a two-way ANOVA (line 202), I was wanting details about the significance of the interaction between the two factors, which is not mentioned in the results. Please could you provide these all details in a standard ANOVA table.

line 223-226: but according to line 222 where the p value is given these differences are non-significant right? In which case, there is no need for them to be highlighted or even mentioned here.

line 226-228: please provide the full statistical details about this negative correlation. A scatter-plot could be provided to visualise the negative correlation.

line 229: I found it difficult to understand what exactly a value of "distribution rate" actually means. It seems from figure 4 that its about the ratio of time spent in the 3 different zones, but how is a single value able to be used for ANOVA derived from the data like this?

line 230-233: again the information line 230 seems to contradict with the information line 232-3. All that needs to be mentioned is whether there was a significant difference for the first contact time between day and night - am I correct in thinking there was no such difference? In this case, the conclusion highlighted in the abstract line 25-26 is incorrect.

line 230-231: Am I correct in thinking that high first contact time is considered here to be synonymous with neophobia? In which case the information on line 230 is basically a repeat to that on line 231. Also, how was "stronger exploratory ability" confirmed, i.e. which specific variable is being referred to here and which of the analyses were done on it.

line 303: I think it is a problem in this manuscript that non-significant patterns are repeatedly being highlighted. There are many different patterns that the reader needs to concentrate on understanding while reading this manuscript, and it is confusing when patterns which are nowhere near significant get mentioned. At first I would think it was important; only later when I realised it was a non-significant pattern I'd then need to disregard it. I urge the authors to please simplify the manuscript by going through and removing all instances of writing about any difference or correlation between treatments which is not significant (P>0.05).

Figure 8: pairwise or post-hoc tests have been done to determine the directions of the differences within these graphs, please could you mention about the pairwise tests in the methods, and describe which specific type of test was done.

line 331: as I mentioned above, I cannot see how a categorical factor like shape can be used in a correlation analysis that requires two continuous variables.

line 375: I think it should be discussed somewhere about how it seems unfeasible that the fish with height of about 6cm (line 103) can even attempt to get inside the reefs with openings of diameter only 3cm.

line 449: This study did not take any measurements associated with attack behaviour.

line 497: I am not sure that a few different types of 30cm acrylic cubes with holes in them really deserves the label of "habitat enrichment technology".

Comments on the Quality of English Language

line 17: change to "reshapes"

line 19-20: change to "affects". Also, on the next line I would change the word "impact" to "affects", because the word impact is normally used to refer to some negative feature, e.g. unwanted environmental damage or something like that.

line 25: change "was" to "were".

line 50-53: this sentence is long and difficult to follow (e.g. I think it needs to be about the behavioural characteristics of fish on artificial reefs, not the behavioural characteristics of the reefs). Please clarify this sentence. 

line 64: change "tool" to "took"

line 77: delete the first "rocky".

line 97: I would change this to something like "These experiments were done in order to"

line 105: I did not understand this sentence.

line 113: change "17 am" to "5 pm"

line 134: change to "aerated"

line 185: delete "preference for"

line 196: please italicise species name.

Figure 4: at top left change "contral" to "control"

line 463-465: this sentence could be written in a more concise way, such as "How effective habitat enrichment is for aquaculture depends on the behaviour of the species involved."

line 470: is it meant to be "ecology of the species" here?

line 492-494: please make this sentence more concise.

Author Response

Thank you very much for your suggestions and comments on our manuscript. These comments have effectively improved our manuscript. We have included almost all of the suggestions, and below, we present our point-by-point responses to the provided comments.

Reviewer 2

  1. line 21: The first abstract sentence is very long and difficult to get specific information from. I would change this to something like: "In order to gain a better understanding of Sparus latus habitat preference and behavioural ecology, we performed an experiment using box-shaped reefs as habitat enrichment materials, allowing us to determine the behavioural strategies and drivers in response to different enrichment structures."

Response to comments: Thank you for your suggestions. We have revised this sentence in the revised manuscript (L22-25).

  1. line 24-25: no significant difference between what?

Response to comments: Thank you for your comments. We have added the following sentence to the revised manuscript: “The results showed that under the different reef opening ratios and shapes, there was no significant difference in reef neophobia of juvenile S. latus” (L25-26).

  1. line 25: to determine that a phobia is specifically for something neo/new, then a more complex experimental design would be needed, e.g. a range of different new things could be provided, and it should be checked if the phobia was for all of them. If only one new thing is provided, then the phobia may be for that specific thing, and if another different thing was newly provided, then maybe there would be no phobia. This certainly cannot be know if only one kind of new thing was provided as part of the experimental design. Overall, it may be better to use a different term here instead of "neophobia". The term "reef taxis" is used a number of times in the manuscript and I would suggest only using this term, as its accurate for what was actually measured in these experiments, more so than the term neophobia.

Response to comments: We agree with your perspective on neophobia experiments. Because the objective of this study was to improve the adaptability to survive neophobia through artificial reef deployment, neophobia is one of the indicators for the degree of adaptation of fish to reefs. Based on the research purpose of determining the impact of reefs, in this manuscript, only reefs are used for experimental verification of neophobia, and the impact of reefs on neophobia of yellowfin sea bream is studied. In future research, we will adopt your suggestion to conduct more in-depth research on neophobia of Sparas latus.

  1. line 27: what is exactly the control group? Is it the experimental treatment where no kind of enrichment structure was provided? Please specify here.

Response to comments: Thank you for your comments. The control group was the environment without artificial reefs, and information on the control group is provided in the revised manuscript (L171 of part 2.4).

  1. line 32: shape is normally a categorical variable, so I would think a correlation analysis (requiring two continuous variables) is not correct here.

Response to comments: Thank you for your comments. We have considered your advice and removed the description in the revised manuscript (L32-34 and L336-344).

  1. line 81: please provide the scientific name for Korean rockfish here.

Response to comments: Thank you for your careful review. We have added the scientific name of Korean rockfish (Sebastes schlegeli) into the revised manuscript (L64-66).

  1. line 96: please clarify the term "abundance pore structure".

Response to comments: Thank you for your comments. We have modified the reef opening structure in the revised manuscript (L105-107).

  1. line 119: what colour was the acrylic?

Response to comments: Thank you for your comments. Both the acrylic plate and reef are grey, as indicated in L134 in the revised manuscript.

  1. line 123: the term "diameter" is only applicable to the circular shape. Possibly it would have been better to calculate the area of the opening and standardise among shapes using that area, rather than the "opening ratio" used here (e.g. the reefs with square openings have the greatest area of opening overall, more than the other treatments, even those with the same "opening ratio").

Response to comments: Thank you for your suggestions. We have changed the opening diameter to the opening height (L132 in the revised manuscript). In this study, the opening height was selected as an index to measure the difference in reef opening, which was mainly divided according to multiples of the body height of the fish. Hence, the opening area was not considered.

  1. line 123: so the reefs with opening diameter of 3cm had openings basically too small for the fish (height about 6cm) to use?

Response to comments: Thank you for your comments. In this study, the height of the experimental fish was 6 cm, and there were 4 opening ratios in the experimental design, namely, 0.5, 1.0, 1.5 and 2 times the body height, which were selected to meet the comprehensiveness of the experimental design. In addition, the reason for designing the experimental group with an opening ratio of 0.5 times the body height was that Sparas latus is a reef-oriented fish, but its reef approach behaviour type remains unclear, including near-reef, contact-reef or entering-reef behaviour. Comparative analysis involving small and large proportions could reveal the reef-feeding type of fish.

  1. line 171: Is it meant here that only these specific shapes/sizes were used? Why not use all the combinations of shapes and sizes as was done in the daytime experiment?

Response to comments: Thank you for your comments. We studied the reef behaviour characteristics of Sparas latus during the day and selected four types of artificial reefs with the best and worst reef tendency behaviours for nighttime experimental analysis. This reduced the time needed for the experiment and demonstrated the tendency of S. latus towards the reef at night.

  1. line 172: it states each experimental set up had three replicates. Please can you confirm here or somewhere else in the methods whether the average parameters of all fish in each replicate were used for the statistical analyses, or whether data from each individual fish within the replicate were used for the analyses..

Response to comments: Thank you for your comments. We have described the statistical methods used for determining the fish observation parameters in the revised manuscript (L190-191). In this study, we randomly selected the average values of the behavioural parameters of 5 fish in each of the three repeated experimental groups for statistical analysis.

  1. line 176: I guess the filming at night was done using infra red light, if so please state that here. And also, it would be important that none of the light is near-infrared, which the fish may be sensitive to. If it cannot be confirmed from past research that the fish are not affected by any near-infrared light of such a camera, then this is something that would need to be tested as part of the current study.

Response to comments: Thank you for your careful assessment. As noted, filming at night was performed using infrared light, and we have added the nighttime filming conditions to the revised manuscript (L184-185).

  1. line 183: but the filming began straight away after the fish were put in the experimental enclosure (line 164) so basically everything in the enclosure, and even the enclosure itself, was a novel object.

Response to comments: Thank you for your comments. In addition to the reef, the experimental pool comprises Sparas latus, so a control group was established in this study to determine the influence of the pool wall on the behaviour of S. latus in the absence of a reef. Joint analysis of these two aspects could reduce the number of experiments for studying the influence of the pool wall on the behaviour of fish.

  1. line 202: The assumption of homogeneity of variances should be tested before ANOVA like this. For testing this assumption, please use Cochran's test or Levene's test. If the assumption is not met, then the data may need to be transformed and it may influence the results and conclusions.

Response to comments: Thank you for your advice. We conducted the Shapiro‒Wilk test and ANOVA before using R for ANOVA; the P values in the three analysis groups were all greater than 0.05, with no significant difference, and the hypothesis of homogeneity of variance could not be rejected. Cochran's test, Levene's test, the Shapiro‒Wilk normality test and ANOVA are required before ANOVA, so they are not specifically proposed in this manuscript. Only the ANOVA results are provided in the Conclusion.

  1. line 218: does contact time mean when a fish physically touches the acrylic reef? I would have thought that some measurement would have been taken about the fish actually going inside the reef. Why was this not measured?

Response to comments: Thank you for your comments. The contact time in this study refers to the time when the experimental fish first touches the reef. There are no measurements of Sparas latus movement within the reef, mainly due to the inability of cameras to penetrate the reef. In future studies, we will consider your suggestions, optimize the test method, and accurately observe and analyse the behavioural characteristics of fish.

  1. line 219: I do not understand why its stated here that different reef structures had an effect, but then the next sentence states "under different reef opening ratios and shapes, there was no significant difference in the time when the experimental fish first came into contact with the reef". These two statements seem to contradict.

Response to comments: Thank you for your suggestions. We have revised this part of the description in the revised manuscript (L230-239).

According to the first contact time between the experimental fish and reef, different reef structures and day and night conditions influenced the neophobic and adventurous behaviour of the experimental fish (Figure 3). In the diamond-shaped reef treatment group, the fish exhibited the strongest adventurous behaviour, with an average time of 245.88 seconds for first contact with the fish reef. Moreover, in the circular reef treatment group, the response time was 513.25 seconds, and the slowest response time was obtained in the square reef treatment group, at 783 seconds. The fear of novelty behaviour of the experimental fish was negatively correlated with the final average distribution rate in the reef area. The shorter the initial exploration time was, the greater the average distribution rate in the reef model area.

  1. line 222: please provide all the details for the ANOVA statistical output (e.g. sums of squares, F values, specific P values). In particular, seeing as this was a two-way ANOVA (line 202), I was wanting details about the significance of the interaction between the two factors, which is not mentioned in the results. Please could you provide these all details in a standard ANOVA table.

Response to comments: Thank you for your comments. A standard ANOVA table is provided in the revised manuscript (L810-815).

  1. line 223-226: but according to line 222 where the p value is given these differences are non-significant right? In which case, there is no need for them to be highlighted or even mentioned here.

Response to comments: Thank you for your comments. We have removed this description and revised this paragraph in the revised manuscript (L230-239).

  1. line 226-228: please provide the full statistical details about this negative correlation. A scatter-plot could be provided to visualise the negative correlation.

Response to comments: Thank you for your suggestions. We have supplemented the analytical methods, scatter plots and descriptions of the results in the revised manuscript (L209-213, L234-239 and Figure 4).

Pearson’s correlation analysis revealed that the fear of novelty behaviour of the experimental fish was negatively correlated with the distribution rate in the reef area (P<0.005).

  1. line 229: I found it difficult to understand what exactly a value of "distribution rate" actually means. It seems from figure 4 that its about the ratio of time spent in the 3 different zones, but how is a single value able to be used for ANOVA derived from the data like this?

Response to comments: Thank you for your comments. We have supplemented the analytical methods, scatter plots and descriptions of the results in the revised manuscript (L212-213, L230-242 and Figure 4).

The distribution rate refers to the distribution ratio of S. latus in the three regions A, B and C. Moreover, the sum of the distribution rates in the three regions is 1, where region A denotes the edge area, region B denotes the core area, and region C denotes the artificial reef areas. The greater the distribution rate is in region C, the stronger the reef-tropism behaviour of S. latus. This section focuses on the correlation between neophobia and the reef-tending rate, so the distribution rate in this manuscript refers to the distribution rate of S. latus in region C. To clarify the expression, the manuscript has been revised.

  1. line 230-233: again the information line 230 seems to contradict with the information.

Response to comments: Thank you for your comments. We have improved this part. In addition, the first contact time of S. latus at night was shorter than that during the day under the corresponding reef shape (Fig. 4c), and at night, neophobia of S. latus was lower, and the exploration behaviour was stronger, indicating that light conditions constitute one of the external conditions affecting the adventurous behaviour of S. latus (L239-242).

  1. line 232-3. All that needs to be mentioned is whether there was a significant difference for the first contact time between day and night - am I correct in thinking there was no such difference? In this case, the conclusion highlighted in the abstract line 25-26 is incorrect.

Response to comments: Thank you for your comments. We have improved this part in the revised manuscript (L25-26 and L239-242). In addition, the first contact time of S. latus at night was shorter than that during the day under the corresponding reef shape (Fig. 4c), and at night, neophobia of S. latus was lower, and the exploration behaviours was stronger, indicating that light conditions are among the external conditions affecting the adventitious behaviour of S. latus.

  1. line 230-231: Am I correct in thinking that high first contact time is considered here to be synonymous with neophobia? In which case the information on line 230 is basically a repeat to that on line 231. Also, how was "stronger exploratory ability" confirmed, i.e. which specific variable is being referred to here and which of the analyses were done on it.

Response to comments: Thank you for your careful assessment. We have improved the analysis description in the revised manuscript (L239-242).

  1. line 303: I think it is a problem in this manuscript that non-significant patterns are repeatedly being highlighted. There are many different patterns that the reader needs to concentrate on understanding while reading this manuscript, and it is confusing when patterns which are nowhere near significant get mentioned. At first I would think it was important; only later when I realised it was a non-significant pattern I'd then need to disregard it. I urge the authors to please simplify the manuscript by going through and removing all instances of writing about any difference or correlation between treatments which is not significant (P>0.05).

Response to comments: Thank you for your suggestions. We have reviewed, deleted and added relevant information to the full text.

  1. Figure 8: pairwise or post-hoc tests have been done to determine the directions of the differences within these graphs, please could you mention about the pairwise tests in the methods, and describe which specific type of test was done.

Response to comments: Thank you for your comments. We have added the appropriate test method to the Methods section. In this study, the F test was used to analyse the difference between the daytime and nighttime movement indicators (L220-222).

  1. line 331: as I mentioned above, I cannot see how a categorical factor like shape can be used in a correlation analysis that requires two continuous variables.

Response to comments: Thank you for your comments. We have considered your advice and removed the description in the revised manuscript (L336-344 and Figure 10).

  1. line 375: I think it should be discussed somewhere about how it seems unfeasible that the fish with height of about 6cm (line 103) can even attempt to get inside the reefs with openings of diameter only 3cm.

Response to comments: Thank you for your comments. We have added this information to the Discussion section in the revised manuscript (L383-389).

  1. line 449: This study did not take any measurements associated with attack behavior.

Response to comments: Thank you for your comments. We have removed the “attack behaviour” from the revised manuscript (L458).

  1. line 497: I am not sure that a few different types of 30cm acrylic cubes with holes in them really deserves the label of "habitat enrichment technology".

Response to comments: Thank you for your comments. Environmental enrichment refers to improving the environment of captive animals. For farmed fish, environmental enrichment mainly comprises physical structures (Näslund and Johnsson, 2016). These structures, including for example, bottom layers such as gravel or pipes to offer shelter, may function simultaneously, for example, as visual stimuli. Although the reef structure in this study is simple and small, the physical environment is altered by the artificial reef to provide shelter conditions for S. latus, so it is also a well-known environmental technology.

In addition, we have been inspired by your suggestions. As a preliminary basic research area, this study provides a reference for the subsequent design of more complex environments close to field conditions through the study of reef characteristics.

  1. line 17: change to "reshapes".

Response to comments: Thank you for your comments. This term has been modified in the revised manuscript (L18).

  1. line 19-20: change to "affects". Also, on the next line I would change the word "impact" to "affects", because the word impact is normally used to refer to some negative feature, e.g. unwanted environmental damage or something like that.

Response to comments: Thank you for your careful review. This term has been modified in the revised manuscript (L19-20).

  1. line 25: change "was" to "were".

Response to comments: Thank you for your comments. This term has been modified in the revised manuscript (L25).

  1. line 50-53: this sentence is long and difficult to follow (e.g. I think it needs to be about the behavioural characteristics of fish on artificial reefs, not the behavioural characteristics of the reefs). Please clarify this sentence.

Response to comments: Thank you for your careful evaluation. This term has been modified in the revised manuscript (L50-53).

  1. line 64: change "tool" to "took".

Response to comments: Thank you for your comments. This term has been modified in the revised manuscript (L80-81).

  1. line 77: delete the first "rocky".

Response to comments: Thank you for your careful assessment. This term has been modified in the revised manuscript (L60).

  1. line 97: I would change this to something like "These experiments were done in order to".

Response to comments: Thank you for your comments. This term has been modified in the revised manuscript (L107).

  1. line 105: I did not understand this sentence.

Response to comments: Thank you for your careful review. This term has been modified in the revised manuscript (L115).

  1. line 113: change "17 am" to "5 pm".

Response to comments: Thank you for your comments. This term has been modified in the revised manuscript (L122).

  1. line 134: change to "aerated".

Response to comments: Thank you for your careful reading. This term has been modified in the revised manuscript (L142-143).

  1. line 185: delete "preference for".

Response to comments: Thank you for your comments. This term has been modified in the revised manuscript (L195).

  1. line 196: please italicise species name.

Response to comments: Thank you for your careful review. This term has been modified in the revised manuscript (L206).

  1. Figure 4: at top left change "contral" to "control".

Response to comments: Thank you for your comments. This term has been modified in the revised manuscript (Figure 5).

  1. line 463-465: this sentence could be written in a more concise way, such as "How effective habitat enrichment is for aquaculture depends on the behaviour of the species involved."

Response to comments: Thank you for your careful review. This term has been modified in the revised manuscript (L469-472).

  1. line 470: is it meant to be "ecology of the species" here?

Response to comments: Thank you for your comments. This term has been modified in the revised manuscript (L471).

  1. line 492-494: please make this sentence more concise.

Response to comments: Thank you for your careful assessment. It has been modified in the revised manuscript (L492-499).

References

Näslund J, Johnsson JI. Environmental enrichment for fish in captive environments: effects of physical structures and substrates. Fish and Fisheries 2016; 17: 1-30.

Round 2

Reviewer 2 Report

Comments and Suggestions for Authors

line 88: please provide the common name here of the study species as well as the scientific name.

line 115: I think the text "and before the experiment in a breeding pond for temporary cultivation." can be deleted as its repeated on the next line.

line 128: I do not understand what is meant by "1:10 ratio", please clarify.

line 146: based on Figure 2, it seems only the four corner areas, and not the other edge areas, were used as Zone A. Please clarify about this.

line 186: as I said in my last review, many fish have eyes that are able to see near infrared light. I did a quick check and the wavelength of light used in typical infrared lamps is 850nm, and this is apparently the same wavelength that can be detected by fish such as carp and zebra fish and others. So even though the night time tanks would look completely dark to us it might have been essentially illuminated for these fish, i.e. not night time. Have there been any experiments on the wavelengths of light that eyes of S. latus can detect? At the very least, it would be good for a caveat to be added to this manuscript about how it may be unknown whether or not this fish species can detect and/or have their behaviour affected by infrared lamps like the one used in this experiment.

line 194: add "in this case, the artificial reef" at the end of the sentence here.

line 212: I said in my last review about the need for testing of the ANOVA assumption of variance homogeneity, but I did not understand the response to what I said in the reply to reviewers text. In that text, the authors stated "We conducted the Shapiro‒Wilk test and ANOVA before using R for ANOVA". What does it mean to conduct ANOVA before using R for ANOVA? (Its good the Shapiro-Wilk test for normality was done before the ANOVA, but this has not been included in the manuscript, which the authors should consider doing). In my previous review I suggested using Levene's test or Cochran's test, and in the reply to reviewer text the authors then said "Cochran's test, Levene's test, the Shapiro‒Wilk normality test and ANOVA are required before ANOVA, so they are not specifically proposed in this manuscript." Overall, it does not seem to me the authors have given any reason why the test of variance homogeneity does not need to be done, but then they state such a test is not proposed in this manuscript. 

line 232: According to the P values provided in Figure 3, none of these differences are statistically significant (e.g. day/night difference p=0.62). So if my understanding is correct, it would be more accurate to say that according to the first contact time between the experimental fish and the reef, different reef structures and day and night conditions did not significantly affect the neophobic and adventitious behaviour of the experimental fish (Figure 3).

line 239-242: maybe the reason they contact the reef sooner at night compared to at day is just because they cannot see where they are going as they swim around, so they randomly bump into anything that is put in the tank, and so the difference between night and day does not really have anything to do with any process of "neophobia". I stated in my last review that proper testing of neophobia must involve some complex experimental methodologies/designs which have not been used in the current manuscript, so it would probably be most accurate to not use here the term neophobia so much (the precise term of "first contact time" could be used).

line 260: did this treatment just have a greater mean, or was the difference statistically significant? Please explain this also for the difference stated on line 264.

line 286-288: please provide the p values for the permutational multivariate analysis of variance, and the ANOSIM, not only the R squared and R values.

line 452-453: But according to Fig. 3c, the ANOVA for this difference had a p value of 0.62, so your study showed there was no significant neophobia day/night difference. 

line 502-503: I would change this to "were explored through testing fish responses to small-scale artificial reef structures in laboratory experiments".

line 513-514: I found it confusing how its stated here about the importance of cultivating farmed fish in conditions that are closer to the natural environment, but the experiments in this manuscript used plastic boxes with holes in them that are unlike any structure naturally found in the natural environment. If the structures need to be like in the natural environment, then naturally shaped rocks or vegetation structures could be used.

Comments on the Quality of English Language

line 16: delete "environment"

line 22-23: delete the text in brackets.

line 86: delete "between individuals"

line 128: change to "five identical grey coloured acrylic plates", and delete the reference to the reef colour line 134.

line 231: is the word here meant to be "adventurous"?

line 337: what is meant here by "major taxonomic groups"?

line 365: delete "underwater"

line 384: I do not understand what is meant by "ratio of reef tending", please clarify.

line 397-399: I did not understand this sentence, please clarify.

line 418-420: I did not understand what was meant by "the nearest neighbour distance of S. latus in the experiment never exceeded the distance between the two individuals", please clarify.

line 455: please italicize species name.

line 469-471: This sentence is still long and difficult to follow even after I made a comment about it in my last review.

line 476: I think the word "specifications" needs to be changed to something else.

Author Response

Thank you very much for your suggestions and comments on our manuscript. These comments have improved the manuscript effectively. We have almost included all of suggestions and below we present a point-by-point response to the comments.

Comments from the reviewers:

  1. line 88: please provide the common name here of the study species as well as the scientific name.

Response to comments: Thanks for your advice. The manuscript has been revised (in the revised manuscript L92).

  1. line 115: I think the text "and before the experiment in a breeding pond for temporary cultivation." can be deleted as its repeated on the next line.

Response to comments: Thank you for your advice. This sentence has been deleted in the revised manuscript (in revised manuscript L123).

  1. line 128: I do not understand what is meant by "1:10 ratio", please clarify.

Response to comments: Thanks for your comments. At present, there are various specifications of artificial reefs launched in the sea area, and the design specifications of relevant artificial reefs participated by this team are generally 3m×3m×3m. The size of the reef in the manuscript is reduced by this, and the reduction ratio is 1:10. To make the article concise, we have changed this sentence to the artificial reef size of 30cm × 30cm × 30cm in the revised manuscript L137.

  1. line 146: based on Figure 2, it seems only the four corner areas, and not the other edge areas, were used as Zone A. Please clarify about this.

Response to comments: Thanks for your careful work. It has been modified in the revised manuscript L161.

  1. line 186: as I said in my last review, many fish have eyes that are able to see near infrared light. I did a quick check and the wavelength of light used in typical infrared lamps is 850nm, and this is apparently the same wavelength that can be detected by fish such as carp and zebra fish and others. So even though the night time tanks would look completely dark to us it might have been essentially illuminated for these fish, i.e. not night time. Have there been any experiments on the wavelengths of light that eyes of S. latus can detect? At the very least, it would be good for a caveat to be added to this manuscript about how it may be unknown whether or not this fish species can detect and/or have their behaviour affected by infrared lamps like the one used in this experiment.

Response to comments: Thanks for your careful work. This was added to the discussion in the revised manuscript L568-574.

In addition, filming at night was performed using infrared light, the wavelength of light used in typical infrared lamps is 850nm, Some fish, such as carp, Nile tilapia(Taro Matsumoto and Kawamura, 2005), and zebrafish(Del Bene et al., 2018), can detect these wavelengths, but the wavelength range of infrared light in S. latus has not been studied. Therefore, it is not known whether the S. latus can detect infrared light and whether their behavior is affected by the infrared light used in this experiment, which will require further study.

  1. line 194: add "in this case, the artificial reef" at the end of the sentence here.

Response to comments: Thanks for your comments. It has been added in the revised manuscript L211.

  1. line 212: I said in my last review about the need for testing of the ANOVA assumption of variance homogeneity, but I did not understand the response to what I said in the reply to reviewers text. In that text, the authors stated "We conducted the Shapiro‒Wilk test and ANOVA before using R for ANOVA". What does it mean to conduct ANOVA before using R for ANOVA? (Its good the Shapiro-Wilk test for normality was done before the ANOVA, but this has not been included in the manuscript, which the authors should consider doing). In my previous review I suggested using Levene's test or Cochran's test, and in the reply to reviewer text the authors then said "Cochran's test, Levene's test, the Shapiro‒Wilk normality test and ANOVA are required before ANOVA, so they are not specifically proposed in this manuscript." Overall, it does not seem to me the authors have given any reason why the test of variance homogeneity does not need to be done, but then they state such a test is not proposed in this manuscript.

Response to comments: Thanks for your comments. The Shapiro-Wilk test for normality was done before the ANOVA, and added the results in the revised manuscript L229-230, L250-252 and Figure 4.

  1. line 232: According to the P values provided in Figure 3, none of these differences are statistically significant (e.g. day/night difference p=0.62). So if my understanding is correct, it would be more accurate to say that according to the first contact time between the experimental fish and the reef, different reef structures and day and night conditions did not significantly affect the neophobic and adventitious behaviour of the experimental fish (Figure 3).

Response to comments: Thanks for your comments. Your understanding is correct. Different reef structures and the length of time the experimental fish first come into contact with the reef under day and night conditions vary, but there is no significant difference between the groups.

  1. line 239-242: maybe the reason they contact the reef sooner at night compared to at day is just because they cannot see where they are going as they swim around, so they randomly bump into anything that is put in the tank, and so the difference between night and day does not really have anything to do with any process of "neophobia". I stated in my last review that proper testing of neophobia must involve some complex experimental methodologies/designs which have not been used in the current manuscript, so it would probably be most accurate to not use here the term neophobia so much (the precise term of "first contact time" could be used).

Response to comments: Thanks for your comments. It has been modified in the revised manuscript.

  1. line 260: did this treatment just have a greater mean, or was the difference statistically significant? Please explain this also for the difference stated on line 264.

Response to comments: Thanks for your comments. It is only shown that the distribution rate has a greater mean, not statistically significant.

  1. line 286-288: please provide the p values for the permutational multivariate analysis of variance, and the ANOSIM, not only the R squared and R values.

Response to comments: Thanks for your comments. It has been modified in the revised manuscript L329-331 and Figure 8.

  1. line 452-453: But according to Fig. 3c, the ANOVA for this difference had a p value of 0.62, so your study showed there was no significant neophobia day/night difference.

Response to comments: Thanks for your comments. Just as you said, different reef structures and the length of time the experimental fish first come into contact with the reef under day and night conditions vary, but there is no significant difference between the groups.

  1. line 502-503: I would change this to "were explored through testing fish responses to small-scale artificial reef structures in laboratory experiments".

Response to comments: Thanks for your careful work. It has been modified in the revised manuscript L577-578.

  1. line 513-514: I found it confusing how its stated here about the importance of cultivating farmed fish in conditions that are closer to the natural environment, but the experiments in this manuscript used plastic boxes with holes in them that are unlike any structure naturally found in the natural environment. If the structures need to be like in the natural environment, then naturally shaped rocks or vegetation structures could be used.

Response to comments: Thanks for your useful suggestions. I can't agree with you more. Experiments with the same structure artificial reefs as those in natural habitats will help to accurately understand the behavior characteristics of Sparus latus, but the reef structure is complex and there are many variable conditions. Therefore, from the perspective of small experiments, this study explores the behavior of fish from a single variable (the shape and size of the opening) to provide basic data for the subsequent research on the behavior characteristics of natural environment.

Comments on the Quality of English Language

  1. line 16: delete "environment"

Response to comments: Thanks for your comments. It has been modified in the revised manuscript L16.

  1. line 22-23: delete the text in brackets.

Response to comments: Thanks for your comments. It has been modified in the revised manuscript L21.

  1. line 86: delete "between individuals"

Response to comments: Thanks for your careful work. It has been modified in the revised manuscript L89.

  1. line 128: change to "five identical grey coloured acrylic plates", and delete the reference to the reef colour line 134.

Response to comments: Thanks for your careful work. It has been modified in the revised manuscript L135 and L142.

  1. line 231: is the word here meant to be "adventurous"?

Response to comments: Thanks for your comments. We have changed it to "First contact time." (L231 in revised manuscript).

  1. line 337: what is meant here by "major taxonomic groups"?

Response to comments: Thanks for your comments. We have deleted it in the revised manuscript L392.

  1. line 365: delete "underwater".

Response to comments: Thanks for your careful work. We have deleted it in the revised manuscript L426.

  1. line 384: I do not understand what is meant by "ratio of reef tending", please clarify.

Response to comments: Thanks for your useful suggestions. We have changed it to "performance of approaching reefs". It indicates the ability of fish to approach the reef in the revision manuscript L445-446.

  1. line 397-399: I did not understand this sentence, please clarify.

Response to comments: Thanks for your comments. We have modified this sentence to "Our study showed that the number of experimental fish in each group was the same, so we failed to reflect the effect of density on fish aggregation behavior" in the revised manuscript L460.

  1. line 418-420: I did not understand what was meant by "the nearest neighbour distance of S. latus in the experiment never exceeded the distance between the two individuals", please clarify.

Response to comments: Thanks for your careful work. We didn't say that clearly, It means the nearest neighbour distance of S. latus in the experiment never exceeded the two body lengths (in the revised manuscript L485).

  1. line 455: please italicize species name.

Response to comments: Thanks for your comments. It has been modified in the revised manuscript L521.

  1. line 469-471: This sentence is still long and difficult to follow even after I made a comment about it in my last review.

Response to comments: Thanks for your comments. It has been modified in the revised manuscript L541-543.

  1. line 476: I think the word "specifications" needs to be changed to something else.

Response to comments: Thanks for your useful suggestions. We have modified the “specifications” to “species” in the revised manuscript L546.
